# Exploring behaviours perceived as important for human—Dog bonding and their translation to a robotic platform

Katie A. Riddoch[1©], Roxanne D. Hawkins[2©], Emily S. Cross[1,3,4©]*

**1** Institute of Neuroscience and Psychology, University of Glasgow, Glasgow, Scotland, United Kingdom,
**2** School of Health in Social Science, University of Edinburgh, Edinburgh, Scotland, United Kingdom,
**3** Department of Cognitive Science, Macquarie University, Sydney, Australia, **4** MARCS Institute for Brain, Behaviour & Development, Western Sydney University, Sydney, Australia

© These authors contributed equally to this work.
\* e.cross@westernsydney.edu.au

**Data Availability Statement:** All data files are available on our Open Science Framework (OSF) webpage - https://osf.io/ycrwh/.

## Abstract

To facilitate long-term engagement with social robots, emerging evidence suggests that modelling robots on social animals with whom many people form enduring social bonds–specifically, pet dogs–may be useful. However, scientific understanding of the features of pet dogs that are important for establishing and maintaining social bonds remains limited to broad qualities that are liked, as opposed to specific behaviours. To better understand dog behaviours that are perceived as important for facilitating social bonds between owner and pet, we surveyed current dog owners (n = 153) with open-ended questions about their dogs' behaviours. Thematic analysis identified 7 categories of behaviours perceived as important to human—dog bonding, including: 1) attunement, 2) communication, 3) consistency and predictability, 4) physical affection, 5) positivity and enthusiasm, 6) proximity, and 7) shared activities. We consider the feasibility of translating these behaviours into a social robotic platform, and signpost potential barriers moving forward. In addition to providing insight into important behaviours for human—dog bonding, this work provides a springboard for those hoping to implement dog behaviours into animal-like artificial agents designed for social roles.

## Introduction

In an attempt to reduce loneliness and improve mental health, a range of technological solutions are being developed and studied, including apps, chatbots, avatars, virtual reality solutions, and robots [1]. Unlike many screen-based technologies, robots are better equipped to physically interact with a person (e.g., via mutual touch) and their surroundings (e.g., by moving objects, or navigating to different parts of a home). It has been suggested that if such robots are paired with social capabilities (e.g., the ability to understand and respond to people's verbal and non-verbal behaviour in an appropriate manner), they could be a candidate solution for helping to combat loneliness and helping to support aging in place [1, 2].

**Funding:** This work has received funding from the European Research Council (ERC) under the European Union's Horizon2020 research and innovation programme (grant agree- ment number ERC-2015-StG-677270-SOCIALROBOTS to ESC), the Leverhulme Trust (PLP-2018-152 to ESC) and an Industrial Strategy PhD studentship from the Economic and Social Research Council (grant number 1945868 to KR and ESC). The funders had no role in study design, data collection and analysis, decision to publish, or preparation of the manuscript.

If autonomous robotic solutions are going to be deployed as a tool to help combat the grow-ing loneliness epidemic, it is vital to ensure the robots tasked with serving such functions look and act the part. Recently, to sidestep the costs and unrealistically high expectations associated with enlisting humanoid robots to serve in this endeavour [3–5], researchers have turned their attention towards modelling socially assistive robots on non-human animals [5–8]. Further-more, due to the success of human-pet relationships, and the relative simplicity of modelling non-human animal forms and behaviours (compared to human forms and behaviour), a num-ber of researchers are beginning to focus more on the mental health benefits associated with pet dog companionship [9, 10] and the implementation of dog behaviours within a robotic platform [11–13].

## Dog-inspired robots

Across many western cultures, pet dogs can provide a source of comfort and companionship, and ownership has been found to benefit a person's mental health, well-being, and ability to deal with trauma [14, 15]. It has been proposed that by modelling social robots on pet dogs (both in appearance and behaviour), end-users could potentially experience similar benefits [11, 16]. Furthermore, social robots that resemble dogs may lead human users to treat these robots more like a companion (as happens with real dogs) than a machine or toy composed of metal and plastic (for some examples, see Fig 1). This, in turn, has the potential to enable users to reap the benefits of robot ownership long term [11]. Robotic dogs equipped with sophisti-cated technology to invite and sustain social engagement would also make the benefits of dog ownership accessible to those who are unable to look after a real dog due to allergies, lack of time, funds, or mental capacity (such as in the case of severe dementia) [17, 18].

However, it remains imperative to also consider the kinds of social and ethical challenges that the development and introduction of robotic dogs as social companions might bring. In a recent scoping review that evaluated nine studies examining the delivery and impact of inter-active pets for older adults (including those with dementia), the authors identified a number of common concerns. These included human users misperceiving robotic pets as living beings (and ethical issues related to deception, even if unintended), ethical issues of attachment, potential negative user reactions, and a number of more practical concerns, such as hygiene and cost [19]. A number of philosophers have weighed in on the moral and ethical challenges

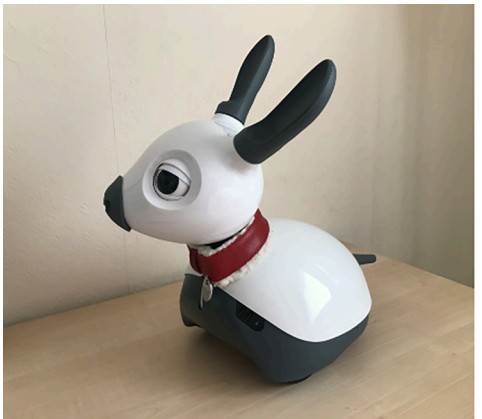 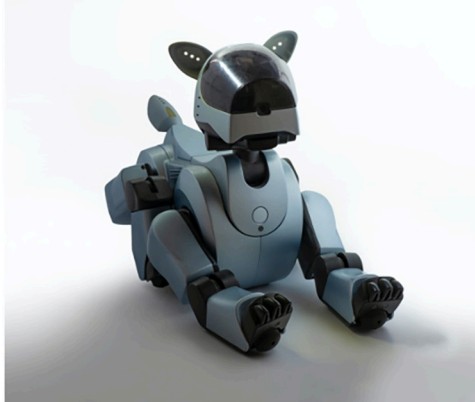

**Fig 1. Images of two robots inspired by dogs, amongst other animals.** The MiRo-E Robotic Platform (Consequential Robotics) and AIBO (Sony). Image credits: Katie Riddoch (author; left image) and Jeremy Bishop via Unsplash (right image).

related to the use of robot pets for social companionship, including Robert Sparrow, who voiced concerns over two decades ago that remain strikingly relevant now [20]. Specifically, Sparrow wrote:

> *"For an individual to benefit significantly from ownership of a robot pet they must systematically delude themselves regarding the real nature of their relation with the animal. It requires sentimentality of a morally deplorable sort. Indulging in such sentimentality violates a (weak) duty that we have to ourselves to apprehend the world accurately. The design and manufacture of these robots is unethical in so far as it presupposes or encourages this delusion,"*
> (p. 305)

In order to understand the scope and limits of robotic dogs as effective social companions, and indeed, explore the extent to which the general public perceives robotic dogs as ethically questionable or acceptable, researchers may wish to model dog behaviours on appropriate robotic platforms and systematically evaluate the efficacy of these behaviours (compared to, for example, living pets). However, before such research can begin in earnest, a vital first requirement is a clear understanding of the features and behaviours that are important to building and maintaining the human—dog bond. Our current understanding of the human—dog bond is limited to physical features and behaviours which are liked, opposed to those that are crucial for the formation and maintenance of a strong social bond. For example, King and colleagues conducted a study with a sample drawn from the Australian public (n = 877), and identified that this sample's "ideal dog" is medium sized, short haired, de-sexed, safe with children, fully housetrained, friendly, obedient and healthy [21]. Participants also wanted a dog to come when called, not to escape from their property, to enjoy being petted and to display affection to their owners. Furthermore, participants in this study identified that the ideal dog should require between 16 and 30 minutes of exercise per day and between 1 and 15 minutes of grooming per week. While this work provides important initial insights into ideal features of dogs, it tells us little about how these individual features or behaviours influence the human–dog bond. More recent work by Konok and colleagues sheds some light on the impact of different qualities by asking the general public which features are *most* liked and disliked about dogs [22].

By asking participants "Do you like dogs?" and to "list 3 characteristics of dogs which influenced your answer most", Konok and colleagues identified the following qualities as the most likable: faithfulness, smartness, friendliness, attachment/devotion, individuality/ personality, unconditional love, kindness, and attentiveness [22]. By asking participants "How can you infer the quality? What does he/she do?", the authors also identified specific behaviours (e.g., "obeys" and "learns quickly") associated with each of the qualities (e.g., smart). Such insights are valuable in terms of understanding why dogs are liked, but it could be the case that qualities which facilitate "liking" are different from those which facilitate the formation of strong bonds. This suggestion comes from human studies which demonstrate that we form different types of relationships with varying strengths (e.g., acquaintances vs close friends) depending on the characteristics of the other person relative to the self [23, 24]. Moving forwards, it would be valuable to better understand if there are behaviours specific to forming strong bonds with a dog, opposed to liking.

If we can identify behaviours that are important to building and maintaining positive human—dog bonds specifically, this should bring us one important step closer to facilitating engaging and meaningful social interactions between humans and robots. As well as improving the user's experience, modifications to robotic systems based on such insights could further improving individuals' mental health and public health in general, while bolstering the longer-

term sustainability of the system and thus the reputation of the company and social robotics community.

## Aims

In this exploratory study, our aim was to identify the domestic dog behaviours that facilitate the formation and maintenance of social bonds between owner and dog. To address limitations of previous studies, we used open questions (as opposed to closed or fixed-choice questions), to encourage participants to be detailed in their descriptions of the dog behaviours (as opposed to stating abstract or broad qualities briefly). We also offered dog owners the opportunity to submit video footage of their dogs–facilitating insights about the physicality of the behaviours they perceived as important to the bond with their dog. To gain insight into behaviours key to attachment formation, we planned to explore the results of those exhibiting high vs low attachment to their pet (as indicated through scores on the Lexington Attachment to Pets Scale [25]). By systematically gathering data on these qualities, we can consider the costs and benefits (and barriers) to implementing pet dog behaviours in robotic systems.

## Method

### Preregistration and ethics

This study and all procedures were approved by the University of Glasgow College of Science & Engineering Ethics Committee (Ethics Number 300190287), and the study procedure was pre-registered on the Open Science Framework prior to data collection (*https://osf.io/ycrwh*). All data supplied in this study were anonymised and will be stored for 10 years, after which time they will be deleted. We report how we determined our sample size, all data exclusions, all manipulations, and all measures in the study [26].

### Participants

In total, the Qualtrics survey platform indicated that 283 individuals started the online questionnaire. All participants were recruited by opportunity and snowball sampling over a one-month period, using advertisements posted on various social media platforms. After the removal of incomplete datasets (that is, those who aborted the study at any point, individuals who left data fields empty, or completely blank surveys), 156 datasets remained. We suspect that there was an issue with the survey platform, as the majority of the incomplete datasets (114 of the 127) contained no data at all.

Following the removal of incorrect responding (answering in terms of multiple pets at once [n = 2] or providing one-word answers [n = 1]), 153 datasets remained. This number exceeds the minimum sample size identified by our original power analysis. Specifically, to detect a medium effect size (Cohen's d = 0.5) between two independent groups, G*Power3.1 indicated a required sample size of 128 participants (using a power of 0.8 and an alpha level of 0.05).

We had intended to compare the results of those who reported high vs. low attachment to their dog (as indicated by scores on the validated Lexington Attachment to Pets Questionnaire [25]). However, as participants reported high levels of attachment on average, across the whole group (M = 4.45 out of a possible 5, SD = 0.50), we determined it would not be relevant or suitable based on our sample to form a "low attachment" group for analyses. Therefore, we analysed our participant sample as one group.

All 153 individuals completed the written aspects of the study (consent, fixed-choice questionnaires, open questions. . .), however only 18 submitted videos of their dogs as supporting

material. As a result, this writeup will focus primarily on verbal descriptions provided by participants, with supplementary screenshots from the videos where possible.

All participants were dog owners, over the age of 18 ($M_{Age}$ = 35.67 years, Range = 21–62), and the majority of participants identified as female (female: n = 146, male: n = 5, non-binary: n = 1, agender: n = 1). The dogs varied in terms of sex, breed, size, and age ($M_{Age}$ = 5.18 years, Range = 3m – 15 yrs), in addition to how they were acquired (e.g., from a breeder [n = 77], rescued [n = 48], friend/family member [n = 23], online/newspaper advertisement [n = 5]). There was also variation in terms of the length of ownership, ranging from 1 month to 15 years (M = 4.39 years, SD = 3.61). A comprehensive demographics table (and the demographics questionnaire) is available on the project's OSF site - https://osf.io/ycrwh.

## Procedure

After gaining ethical approval, researchers posted the study advertisement (containing a link to the survey) on various social media platforms. Participants followed the link to the online Qualtrics survey platform (www.qualtrics.com), where they had the opportunity to read an information sheet about the study (written in English). Participants were then asked if they were happy to provide their informed consent. After providing written informed consent, participants completed the Lexington Attachment to Pets Scale (LAPS)– 23 five-point items scored from "Strongly Agree to Strongly Disagree", developed to measure emotional attachment towards a pet [24]. Example items include "I think my pet is just a pet" and "My pet and I have a very close relationship". A full list of items can be found in the S1 File and on the study OSF site (osf.io/ycrwh). Participants were then asked to provide demographic details (e.g., their age, gender, length of dog ownership, dog breed, etc). A full list of demographic questions is available in S2 File and on the study OSF site (osf.io/ycrwh). Participants were then asked to describe behaviours of their dog according to the following instruction:

*"To aid our understanding of human—dog attachment, please describe things that your dog does that you really like. Specifically, behaviours that you think are crucial to the bond you have with your dog. Please describe the behaviour/s in as much detail as possible."*

After providing a written description, participants were given the option to also provide a short video clip showing them and their dog engaging in the behaviour. In addition to being asked if they were happy to provide consent for the research team to access to their video, participants also had the opportunity to provide consent for their uploaded videos to be used in publications and presentations. After providing consent, participants' videos were sent via the GDPR-compliant Glasgow University Transfer System to be securely stored within OneDrive. Finally, participants were given the option to provide an email address to be entered into a prize draw to win one of five £25 Amazon gift cards. 104 individuals submitted their email address, and a random number generator was used to choose the five winners.

## Qualitative data analysis

The open question responses were analysed using Thematic Analysis, following a rigorous six-step method [27]. This is a widely used inductive method of qualitative analysis that involves familiarisation with the data, followed by classification of recurring ideas into codes. These codes are grouped into broader themes, which are then discussed by independent coders. In further screening rounds, the themes are reviewed and refined, and named and renamed, where suitable. The coding process is interactive and incorporates disassembling and reassembling data, evaluation, interpretation, and attempting to draw conclusions [28, 29].

In this study, two coders (KR and RH) undertook the analysis- one analysing the full data-set, and the other analysing a randomly selected subset of the transcripts [20%, n = 31]). After coding the full dataset within NVivo software (v.12), the first coder identified 8 prominent themes. Upon coding the subset of the data, the second coder identified 9 themes. Following verbal discussion of the similarities and differences, the coders agreed on the 7 data-driven themes described in the next section. See S1 Table for details of the themes identified by individual coders, and the resulting convergent themes. Finally, the "Word Search" tool in NVivo was used–allowing us to determine the exact number of participants who mentioned key words associated with each of the themes (See S2 Table). Note–the results of the word search tool were checked manually, to ensure the word was being used in the relevant context.

We had originally planned to explore to results of individuals with high vs low general emotional attachment to their dogs–as indicated by the LAPS questionnaire results [25]. This was not possible, however, as scores on the questionnaire were close to ceiling (on a scale from 1–5, M = 4.45 and SD = 0.50)—indicating high levels of general emotional attachment across the group.

## Results

The themes identified in the data analysis are discussed in the following section, in order of highest to lowest data coverage (See Fig 2 for illustration of the themes and their respective data coverage). There will also be discussion of relevant sub themes (See Table 1 for a summary of themes, sub-themes, and related perceptions–again, in order of data coverage from most to least documented). Screenshots of the video-submissions are included, where possible, to offer insight into the physicality of the dog behaviours discussed.

### Theme 1: Shared activities

A majority of owners mentioned the importance of playing behaviours for the human—dog bond (n = 100, 65%). Importantly, the play behaviours seem to vary—encompassing games

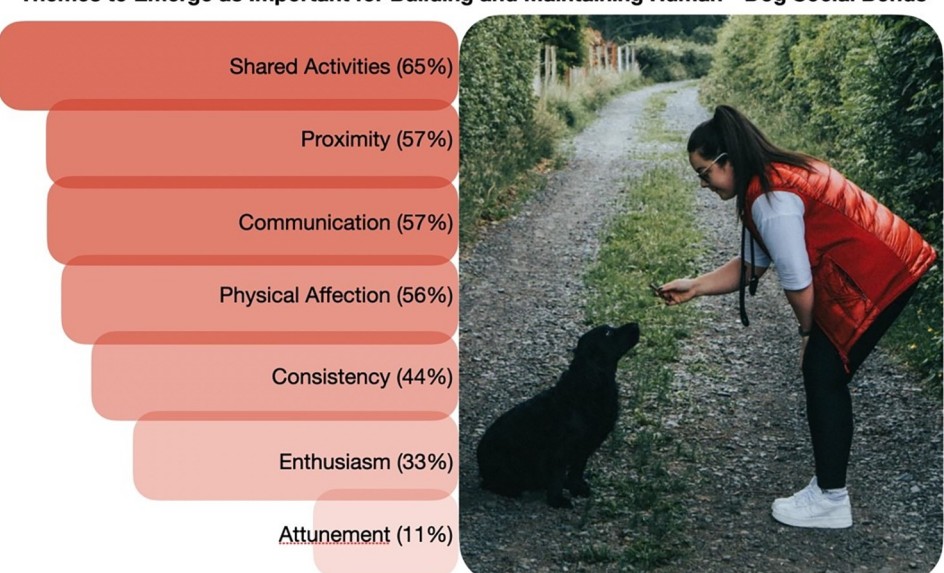

**Fig 2. Visualisation of the themes identified through thematic analysis, with reference to data.** Photo credit: Original photo by Kyle Mackie on Unsplash, reprinted here under a CC BY license.

Table 1. Overview of key themes identified in the study, and related perceptions of dog behaviours.

| Key Themes | Relevant Behaviours | Related perception of the owner, about the dog behaviour |
|---|---|---|
| Shared Activities | • Importance of playing with the owner<br>• Dog cooperates during activities such as walks, visiting the pub<br>• Training | • Mutual liking of certain activities<br>• Mutual enjoyment of activities<br>Shared experiences |
| Proximity | • Follows owner around a space sometimes<br>• Sometimes looks back at the owner, whilst walking ahead<br>• Remains in close proximity whilst on walks<br>• Sleeps in close proximity to the owner–lying next to them, touching them | • Wants to be close to the owner<br>• Owner as safe base for dog<br>• Cares about the owner<br>• Protective of the owner |
| Communication | • Mutual eye-contact: looking at the owner whilst in close proximity, or glancing back at owner when walking ahead<br>• Bringing toys/objects to the owner<br>• Nudging the owner with head or paw.<br>• Responds to owner's voice commands | • Intentional (trying to communicate needs/wants)<br>• Intelligence<br>• Choice or preference for certain activities<br>• Monitoring, checking-in, or protecting the owner<br>• Seeking, paying attention, and listening to the owner |
| Attunement | • Ability to pick up on emotional cues and provides a response'. Not just sitting with them but giving them physical affection as a response to their neg emotions<br>• Recognises time, e.g., when owner is due home, when work is over, when it is bedtime, and joins owner in their routine | • Cares for the owner<br>• Emotionally intelligent<br>• Wants to comfort the owner<br>• Aware of surroundings<br>• Intelligent<br>• Part of the family |
| Physical Touch | • Initiates physical closeness and touch<br>• Joins the owner on the sofa and lie down touching them<br>• Rests or puts head/paw on the owner<br>• Provide physical affection (e.g., kissing)<br>• "Cuddling"<br>• Spontaneous demonstration of behaviours (not always prompted by the owner). | • Wants to be close to the owner<br>• Enjoys spending time with them<br>• Feels comfortable with the owner<br>• Intentional<br>• Cares for the owner<br>• Loves owner<br>• Owner feels loves/wanted/needed |
| Consistency | • Consistently responds/ returns to owner when name is called<br>• Generally obedient in response to words or gestures<br>• Always approaches and greets when owner enters the space/home | • Owner is being listened to<br>• Loyal<br>• Intelligent<br>• Respectful of the owner's wants<br>• Excited to see the owner<br>• Cares about them |
| Positivity and Enthusiasm | • Approaches quickly, jumps up and down, and gives 'kisses' when owner arrives home<br>• Expressing positive facial and bodily expressions | • Excited to see the owner<br>• Cares about them<br>• Smiling or enjoying activities<br>• Consistently positive and up for playing<br>• Appreciates owner |
| Attunement | • Ability to pick up on emotional cues and provides a response'. Not just sitting with them but giving them physical affection as a response to their neg emotions<br>• Recognises time, e.g., when owner is due home, when work is over, when it is bedtime, and joins owner in their routine | • Cares for the owner<br>• Emotionally intelligent<br>• Wants to comfort the owner<br>• Aware of surroundings<br>• Intelligent<br>• Part of the family |

(e.g., tug of war), training through play (e.g., fetch), independent play (e.g., the dog alone, playing with a ball), or the dog exhibiting playful behaviours. See Fig 3 for image of dog and owner engaging in play, followed by independent play.

> "My dog loves to play with me, we enjoy fetch, tug of war, hide and seek and just chasing each other around the house, also some rough and tumble."

> "She likes to skid around on her front paws, which I think is my favourite behaviour of hers, she's a funny girl."

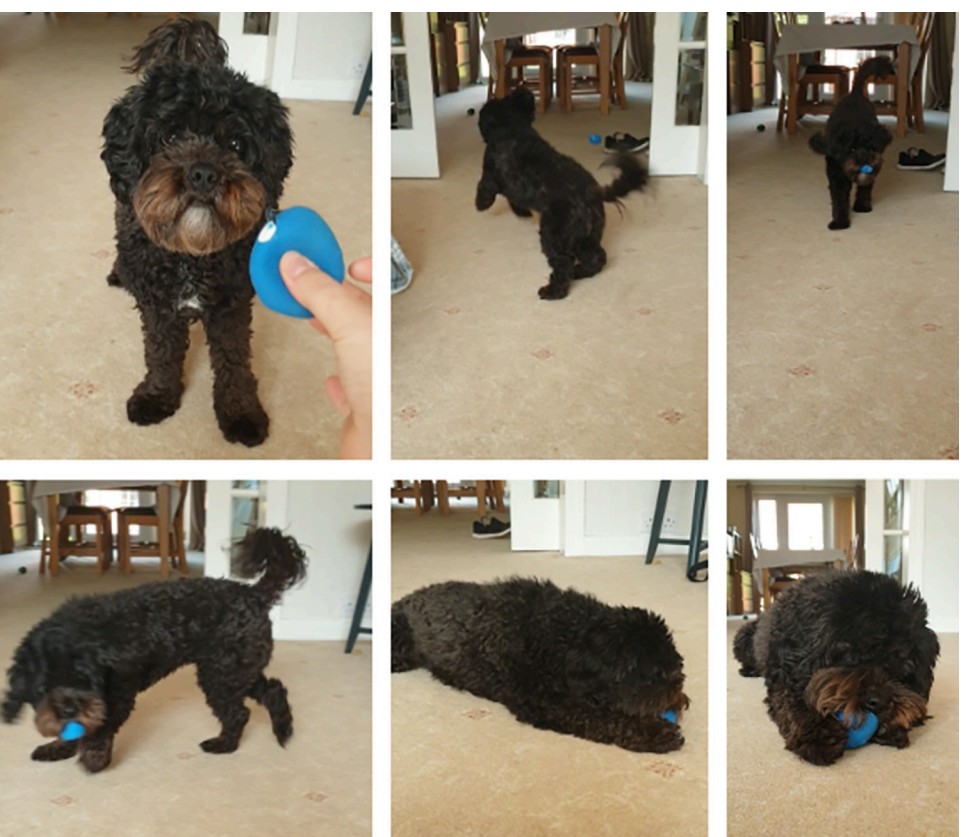

**Fig 3. Screenshots of fetching behaviour, followed by independent play—taken from dog owner video submission.**

There is also mention of play with other family members—as a communal activity that brings the family together. Also, in addition to the act of playing itself, the act of the dog initiating the play is perceived as important, signalling that the dog is excited to play, or to spend time together:

> "If we are playing with him he always shares the play. If I throw the ball, he brings it back to my wife; if she throws it, he brings it back to me."

> "Every evening she likes to play, she'll bark to get our attention and play bow until we start throwing her toys for her."

Repeatedly mentioned as important to the human—dog bond are behaviours that occur whilst on a walk together. Furthermore, the act of walking together was rated as important, as were demonstrations of affection and mutual enjoyment. Additionally, multiple references to the importance of the dog being aware of their owner whilst on the walk were made (such as monitoring behaviours; n = 18, 11%):

> "Every so often when we're walking down the street, she'll just stop in front of me, get up on her hind legs and give me a hug."

> "When going out for a walk, he will always "check in" with me either by looking at me or licking my hand quickly. It is as if he is reminding me that he knows I'm still there and he hasn't forgotten about me even though he is joyfully exploring."

Alongside playing and walking, other shared activities included training, relaxing on the sofa, and additional everyday activities. The dog's apparent mutual enjoyment of shared activities was repeatedly mentioned as important:

*"He loves doing dog agility together, in particular jumping over bars and weaving following my hand and commands."*

*". . .enjoys the same things I do (walking, hiking, swimming, being outdoors)"*

## Theme 2: Proximity

For this theme, proximity refers to the physical distance between owner and dog. Here we found that the majority of participants (n = 88, 57%) mentioned the importance of physical closeness between dog and owner. Numerous owners mentioned specific examples—these included 1) the importance of their dog physically following them around their space (e.g., from room to room; n = 22, 14%) and 2) the dog remaining in close proximity when on walks or within the home (n = 18, 11%). The owners reported perceiving this behaviour as resulting from love, loyalty, or the owner being a perceived source of nurturance or protection:

*"They look for cuddles they follow me around the house from room to room. They are always close by to me"*

*"He never runs too far ahead when off lead and always runs back to me to check in."*

*"The biggest thing for me is the love she shows my children especially the youngest. Luna [the dog] will cry for them, she will sleep in each of their rooms for a short time each and every night before settling in the hall where she can monitor everyone at once."*

Numerous owners (n = 48, 31%) made reference to the importance of sleeping behaviours to the bond between them and their dog. The comments include references to night-time sleeping, napping (short periods of sleep), the dog napping on the person, and co-sleeping (both human and dog sleeping). The locations most frequently mentioned were the sofa/couch and the bed. The importance of physically touching whilst sleeping is repeatedly mentioned. See Fig 4. for image of dogs in close proximity, submitted by dog owners in the study.

*"She sleeps in the bed with myself and my husband and she often naps with me. She likes to be touching one of us when sleeping, so she will rest a paw or her head on one of us."*

*"My dog always sleeps right against my body. He will make sure to have some part of his body touching mine and it makes me feel like he loves me and feels safe with me."*

## Theme 3: Communication

This theme captures the importance of perceived shared communication between owner and dog for the human—dog bond (n = 88, 57%). Numerous owners (n = 27, 16%) referenced the importance of eye gaze. Some dog owners mentioned liking how their dog looks in their eyes, or gazes toward an object and back at them, to indicate their wants or needs (e.g., to play or for food). In contrast, others perceived the gazing behaviours as the dog being attentive to them, and owners indicated that they feel "listened to" as a result. This eye gaze was mentioned as important in training scenarios, but also more generally (e.g., when close to each other, or on walks). See Fig 5. For illustration of gazing behaviours, submitted by dog owners.

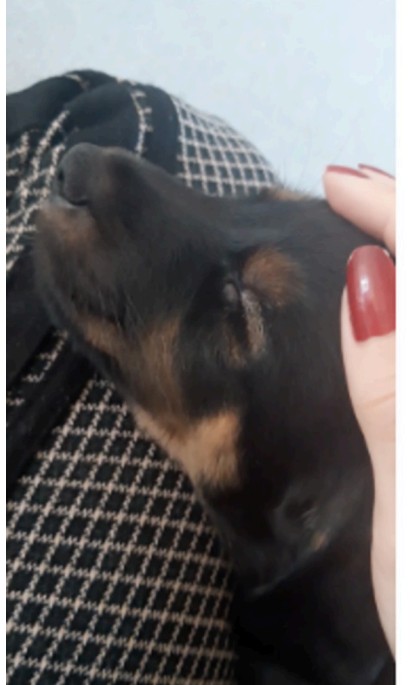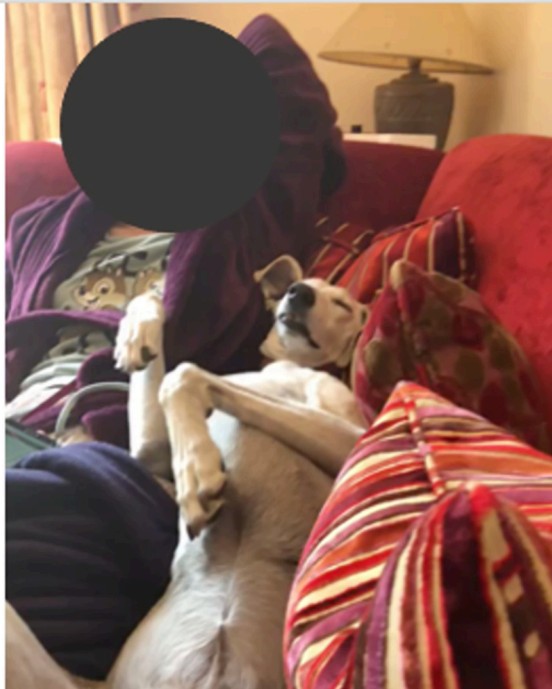

**Fig 4. Screenshots of dogs sleeping in close proximity to their owners, from the dog owner video submissions.**

> "My dog knows when I've finished work and will come and join me in the kitchen, he barks at me/us for attention and if he doesn't get it, he'll toss a ball into the kitchen and just look at it like 'throw it for me please'."

> "I love when he gazes up at me and tilts his head when I talk as if he understands or is listening to what I'm saying."

In addition to the perception of the dog being aware and listening, some owners attributed a deeper meaning to their dog's gaze. Specifically, owners spoke of how they feel their dog looks at them to communicate a strong connection or love for them.

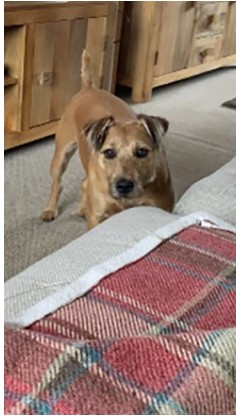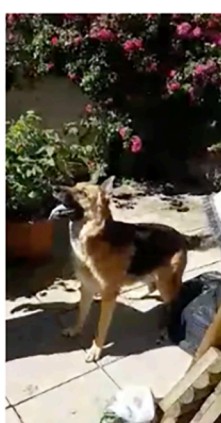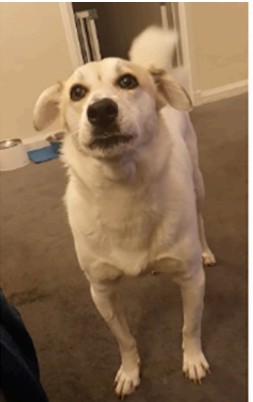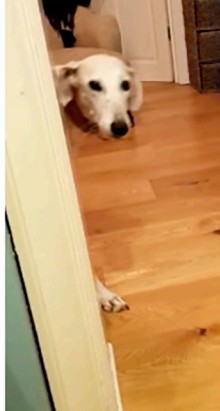

**Fig 5. Screenshot of dogs gazing at owner, from dog owner video submissions.**

*"His eyes and the way he looks into my eyes has always felt very surreal. It truly feels that he is looking at me with intent and trying to communicate his feelings with me."*

*"My dog also looks me directly in the eye. To me, the eye contact symbolizes trust and his affinity to me. . .My dog knows exactly what I need/what from her just by an exchange in looks and vice versa."*

In addition to using eye gaze, some owners (n = 15, 9%) mentioned that they liked how their dog brings toys to them. The purpose of such actions was perceived differently across owners. For example, some owners interpreted this as the dog asking to receive attention or initiate play, others saw this as a form of comfort from dog to owner, and others as the toys being presented as "gifts" for the owner. Across all cases, toy-presenting was perceived as the dog trying to communicate their wants or needs. In all examples, it is implied that the toy-presenting behaviour is spontaneous, rather than being prompted by the owner:

*"He brings me toys to play with him and gets so excited about it."*

*"My dog brings me gifts when I come home or in the morning to wake me up."*

Nudging behaviours were also perceived to be important (n = 7, 4%). The presentation of the nudging differed—touching the owner with the paw, head, or nose, and the behaviour is sometimes described as gentle, but often the intensity is not mentioned. Additionally, some participants report how their dog appears to convey their wants or needs by vocalising (n = 7, 4%). As with toy bringing behaviours, the nudging and noises are perceived as a purposeful behaviour—e.g., the dog wanting attention or affection:

*"I like when he wants clapped and comes and nudges my arm to clap him and he sits up on my knee for a cuddle."*

*"Brings me toys to play with, places head on my lap, paws at my hand for strokes, nudges my elbow to get my attention."*

*"Makes specific noises for different reasons—different ones for pet me, play with me, I need out to the loo"*

A final form of communication was responsiveness to the words or commands of the owner, facilitating feelings of being listened to (n = 15, 9%). Owners also implied that these behaviours were intelligent, and an indicator of their dog's loyalty to them:

*"With me, when I call her back, she comes running really fast and happily, where she does this less with other people."*

*"When I say his name, he wags his tail. He comes to me when I call him. He recognised my voice"*

Related to this theme of Communication, numerous owners explicitly mentioned that they perceive their dog's communicative behaviours as intentional (that is, the dog is exhibiting the behaviour for a specific purpose; n = 28, 18%). Linking to this, participants repeatedly mentioned that they perceive their dog to be "intelligent", or "smart" (n = 26, 16%)–

*"She has learned to manipulate us to get what she wants e.g., treats.".*

*"Every morning and evening after he's eaten his breakfast and dinner he jumps up on the couch and gives me a little lick and it's like he's saying, 'thank you for dinner'."*

*"She learns everything super-fast and has actually taken things like toys upstairs and thrown them out of my bedroom window just so she can watch them fall."*

### Theme 4: Physical touch

Another prominent theme across dog owners (n = 86, 56%) was physical affection. Specifically, participants mentioned their dog engaging in such behaviours as resting their head against the owner, putting a paw on the owner, lying on the owner's feet, lying next to the owner with a body part touching them, giving the owner "kisses" and "hugs", and engaging in cuddles.

*"He will curl up beside me if I take a nap and will lie on top of my lap when I'm watching tv. He climbs up me and gives me kisses and lies on top of me."*

*"She comes to my face and licks my chin then jumps on the bed and cuddles in until I am awake."*

"Cuddling" is repeatedly mentioned as important to the bond (n = 56, 36%), however the descriptions of this behaviour across participants varied. Some accounts referred to cuddling as the dog resting their head upon the owner, whereas others specified that the dog fully lay down next to the owner for sleeping. The majority of mentions provided no detail about the specifics of the cuddling behaviour. It would be insightful to conduct further research to better understand what cuddling consists of, and individual variation in cuddling behaviours and preferences. Although cuddling behaviour is not well understood, there is a consistent perception that the dog is perceived as "enjoying" or "loving" the experience:

*"She absolutely loves cuddling on the sofa and having a nap together during the daytime and I'll sit down first and arrange the cushions and myself so that she can just lie alongside me, and she sticks her nose underneath my arms or in the crook of my neck and we'll just lay there for ages."*

*"Every evening, he snuggles up to me or my husband and places his head in my lap and curls his paw around my arm, almost as if he's cuddling back. He's so gentle if he jumps up, almost knowing that he doesn't want to hurt us."*

Crucially, there were repeated mentions (n = 85, 55%) of how physical touch and affection was spontaneous and dog-initiated (as opposed to being prompted by the owner); these behaviours were perceived as intentional and as a result of love or caring for the owner.

*"He will rub his head against our hands to get a scratch or a pet. All of this makes us feel that the affection we feel for him is reciprocated, which is a pretty amazing and loving feeling."*

*"She is very loving—she will "ask" to sit beside me on the sofa or bed at any opportunity. When I let her up, she snuggles close to me.*

### Theme 5: Consistency

The fifth theme to emerge as important for building and maintaining the human—dog bond concerns the consistency of the behaviours mentioned in the previous sections (n = 68, 44%).

Specifically, participants made recurring mention of the consistently positive nature of the dog, such as expressing positive emotions and excitement to see and be with the owner, and perceived happiness and enjoyment displayed by the dog when in the presence of their owner. Also mentioned is the consistency of the greeting behaviours—that they occur every time the owner arrives home, no matter how long the person had been gone for. As a result of the attentive and consistent enthusiasm, some owners reported feeling loved by their dog.

> *"He always greets us even if we leave the room for five minutes and is very excited to see us."*

> *"He is always so happy to see you come home regardless of how long you were away. He wags his tail and stares at you if you have stopped patting him and he wants more."*

There is also mention of the importance of consistency in terms of listening to the owner, and obeying commands. Particularly important is the dog recalling when the owner calls their name.

> *"My dog follows me around and always listens to me"*

> *"When out walks she never goes too far away from us and always answers to her recall."*

Although consistency is crucial in terms of enthusiasm, positivity, obedience, and emotional awareness, it is implied that inconsistency is also desirable sometimes (n = 20, 13%). For example, through behavioural variability, the owner perceives their dog as having some independence and choice in terms of what they want to do:

> *"He is obedient to a degree but has a mind of his own too? So, he will bring the ball back to us when we ask but maybe not give it to us."*

> *"She often does what she wants, regardless of what I tell her, but she does what's she's told when it matters."*

## Theme 6: Positivity and enthusiasm

This theme captures the importance of the dog's enthusiasm, and the owner's perception that the dog is experiencing happiness and enjoyment. Specifically, one third of the owners (n = 51, 33%) mentioned that when they arrive home, their dog approaches them at the door and does one (or a combination) of the following behaviours: erratic tail wagging ("acts like a propellor", "wags his tail like crazy"), wiggling of the backside, leaping into the person's arms, or jumping up and around "excitedly" and with "delight". The intensity of the dog's excitement was also repeatedly mentioned—"he gets so excited to see me and wags his tail like crazy."

> *"The way she gets so excited when we arrive home, bouncing around the house and not leaving my side, shows me she loves our company. If I stop stroking her, she'll push her head into my hand and nudge it. Her tail acts like a propeller as she's so happy we've returned home."*

> *"She loves to give cuddles and brings me her toys as a present whenever I come home!"*

Numerous owners mentioned that they perceive their dog as funny/silly (n = 13, 8%), and they really like this about them–e.g., "He can be a bit of a silly cookie too when he gets excited or starts rolling around in the leaves or chasing a feather all over, which always gives us a bit of a laugh together, which is nice.". Repeatedly mentioned is also the perception that the dog is

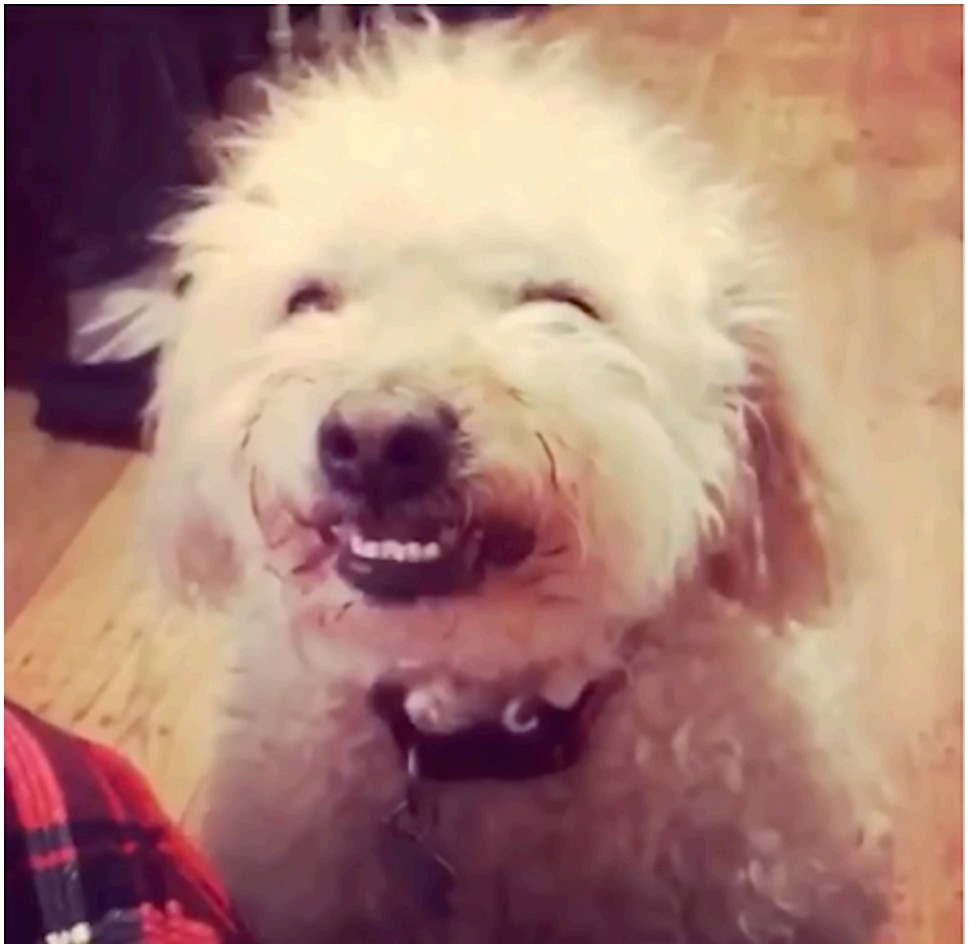

**Fig 6. Screenshot of video from dog owner.** Illustrates an expression that the owner perceives as "smiling".

enjoying the experience with the owner (n = 14, 9%)–e.g., "She enjoys our interactions, and she is happy when I play games with her". Related to the latter, numerous owners express the importance of the dog seemingly "smiling", and the dog's face as a signal of enjoyment. See Fig 6 for images from owners—illustrating the attentiveness of their dog upon arrival home, and their dog "smiling".

> *"When we go for a walk, my dog will often look up at me with just such love in his eyes and a big smile on his face as if he is thanking me and telling me that he is having so much fun"*

> *"He smiles when he's happy like he's letting me know that he's enjoying himself."*

### Theme 7 Attunement

Attunement refers to the ability to sense and respond to another person–for example, picking up on cues that a partner is unhappy (e.g., through facial, verbal, or non-verbal cues) and "being there for them" as a result [30]. In this theme, we discuss the apparent importance of the dog appearing attuned to the owner's routines and emotional states.

Numerous owners (n = 17, 11%) mentioned how their dogs seem to have an awareness of the owner's daily routine (patterns of behaviour that correspond to certain times of the day),

and that they will alter their behaviour in response. For example, the dog will go to the owner around the time that work is done, and they will go lie in the bed just prior to when the owner is due to wake up. There is repeated mention of synchronised sleeping behaviours, which are discussed more thoroughly in "Theme 1: Shared Activities"

> ". . .He also waits on the bed upstairs when we are out and as our pattern is very similar, he somehow knows when we're returning as we have watched him sitting up looking out the window for us minutes beforehand like clockwork (as we have watched him on the indoor cctv). My 7-year-old niece asked, 'how does he know the time?' as he's always waiting at home time."

> "Her time keeping, if I am a minute behind schedule, she will put her paws on my leg to remind me of walks."

The importance of the dog being aware of the owner's work routine, and awaiting their arrival home, was also repeatedly mentioned (n = 17, 11%).

> "Before the last six months [of COVID-related working from home], he would be sitting looking out the window every night waiting on me coming home from work."

> "My dog knows when I've finished work and will come and join me in the kitchen. . ."

It was repeatedly mentioned (n = 37; 22%) that the dog behaves differently around an owner who is upset or stressed, demonstrating the perceived ability to pick up on and respond to owner emotions. In response to the owner being sad or upset, the dog approaches the owner, and will often engage in one of the following behaviours—seek proximity, lie on them, sit next to them, lick their face, provide "kisses", or present toys. In addition to being attuned to the emotions of the owner, the dog is also perceived as being mindful of the emotions of others. The dog is perceived as kind or caring for the family more generally.

> "When either of us are upset she will know and comfort us by lying on us or giving us kisses."

> "She also knows when I'm feeling sad and will stay closer to me when I'm feeling like this. When I cry, she sometimes gets up close to me and licks my face. This feels like she's comforting me."

Participants specified that the dog abstains from normal actions or activities when their owner is experiencing negative emotions.

> "She knows when to be by my side when I'm sad and has an understanding if I'm not mentally feeling up to leave the house, she won't pester me to go for a walk she will be content with playing in the garden for her exercise."

> "Sometimes he seems to know when I'm sad as he'll just come and sit really close to me but doesn't paw or tap for attention.

## Discussion

By using open-ended questions, we gathered rich detail about specific dog behaviours that roughly aligned with seven core themes, which owners perceived as important for establishing and maintaining human—dog bonds. These themes mirror factors important for the

formation and maintenance of human interpersonal relationships identified in previous research literature, including (1) emotional attunement [30]; (2) shared communication [31, 32]; (3) consistency [33, 34]; (4) touch [35]; (5) positivity and enthusiasm [36, 37]; (6) proximity [38, 39]; and (7) mutual enjoyment of shared activities [40, 41]. As similar major themes arise when exploring the development and maintenance of human—human and human—dog close social bonds, the results from this study enhance our understanding of the depth and quality of relationships that can develop between humans and non-human agents. Before we explore these themes and their implications in more detail, we first wish to draw the reader's attention to limitations of the current study, in order to most appropriately and fairly contextualise our findings.

### Study limitations

Through the use of open questions, this study was able to gather detailed and novel insights regarding dog behaviours perceived as important to human—dog attachment. Due to the lack of video submissions however, there remains a lack of understanding with regards to what the specific behaviours look like, and how they differ between individual dogs, and the different human—dog partnerships. This gap in our knowledge is problematic for those developing animal inspired robots, as it means that we cannot accurately visualise the dog behaviours, nor the physical and technical capabilities are required to facilitate them. To close the gaps in our knowledge and aid the development of long-term robotic pets and similar, future work should prioritise capturing the nuances of real dog behaviours, and user insights regarding desirable behavioural boundaries.

Another limitation of the study is that despite conducting this study on a social media platform open to all, the vast majority of respondents identified as female (96%). Although such bias is common in the field of Human—Animal Interaction (HAI) work [42], it is a significant issue which negatively impacts the generalisability of the findings. To avoid this problem moving forwards, future studies should tailor their recruitment methods to encourage a diverse and representative sample–for example, by thoughtfully considering the word choice and imagery used in advertisements, and by posting on a range of social media groups (opposed to those dominated by one group). One might also consider monitoring the demographics of incoming responses, and changing tact during the course of recruitment to create improved balance within the sample.

In this research, a second limitation exists with regards to the generalisability of the results–this time however, by design. Specifically, that all respondents were all individuals who own dogs. On the one hand, this design allowed us to gain insights with depth and breadth, from individuals with extensive experience with dogs–valuable for understanding which behaviours apparently facilitate attachment. On the other hand, this bias means that our insights reflect only the perceptions and preferences of dog owners. In a world where robotic animals will be used by the general public (not just dog owners) this is potentially problematic. By modelling robotic animals on the dog behaviours outlined in this study, developers may create robots which are well-liked by dog owners, but not necessarily the broader public (e.g., individuals who own other types of pets, or people who do not have any experience with animals). To improve the generalisability of the results, and potentially create robotic animals which are accepted on a wide scale, future work should strive to include individuals with a broad range of experience with animals. On a similar thread, to create robots which are widely used, there should also be increased effort to conduct work examining how context, culture, and form, might influence perceptions of such robots [43–46].

## Variation in human—dog relationships

The majority of participants in the present study reported high levels of attachment to their dog (high scores on the Lexington Attachment to Pets Scale [25]). The questionnaire does not, however, offer insight the *nature* of the attachment between the owner and their dog–e.g., secure or insecure, and anxious or avoidant [46]. As a result, it is not possible to tease apart whether different groups (e.g., securely attached vs insecurely attached) hold distinct behaviour preferences in dogs. This comparison is particularly relevant in the context of translating dog behaviours into a robot, as valid concerns exist that some individuals could become overly dependent on robots—to the extent they might neglect their human social relationships and lose their own independence [47, 48]. Through a better understanding of which dog behaviours are associated with different attachment styles, it may be possible to encourage healthy social relationships (opposed to those associated with negative consequences) between humans and robotic dogs.

Moving forwards, it would also be valuable to determine the extent to which other demographics influence people's preferences for robotic dog features and behaviours. For example, both personality type [49] and length of ownership [50] have the potential to further influence wants and needs from a robot dog. Insights about such factors would allow developers to better tailor robotic animals to suit different demographics (e.g., children vs adults) and different purposes (e.g., children in short vs long-term hospital stays). While the current study is not sufficiently powered to conduct comparisons between individual demographic groups or to further investigate the impact of individual differences on dog behaviour preferences, one particularly rich avenue for further research will be to investigate these impacts more closely with appropriately powered research designs.

## Translating dog behaviours onto robotic systems

In this study, the majority of owners made reference to their dog's behaviour as both intelligent and intentional, and ascribed considerable importance to both attributes in terms of bond-building. When these insights are considered in light of how best to develop robotic dogs as social companions, it might at first seem difficult to reconcile how these behaviours could apply to an autonomous machine, whose behaviour might not appear genuinely intelligent or intentional. We would argue, however, that the origin or authenticity of the behaviour is not necessarily important–it is instead the perception of the user that should be the focus of the conversation (e.g., [51, 52]). Specifically, although a robotic dog's behaviour might be controlled by simple algorithms, its actions could still be perceived as intentional or intelligent. This suggestion is supported by a number of experimental studies, which demonstrates that people often adopt an "intentional stance" (opposed to a mechanistic one) when interacting with a robot [53, 54], and people's prior experiences, beliefs and expectations about a robotic system can further up or down-regulate their propensity to attribute social or mind-like attributes to an individual robot (e.g., [52, 54, 55].

On the topic of intention attribution, the related ethical and moral issues will become increasingly important and urgent to consider. Specifically, a number of researchers have raised concerns that by creating the illusion of intelligence or intentionality, people are being deceived by technology [56, 57]. Ethically, this is proposed to be especially problematic when it comes to the deception of vulnerable people, such as children, individuals with additional needs, and people suffering from dementia [56, 58]. Moving forwards, when designing behaviours for a robotic dog, it will be necessary to consider such issues, weigh up the benefits and risks of robot deception, involve researchers from a broad range of disciplines to contribute to design and regulation (including law, ethics, social sciences, developmental psychology, etc),

and be mindful of the state of public opinion at that time of creation and deployment [48, 57, 59].

Returning to the present results, we also found that the majority of participants (>50%) identified the following as important to the bond with their dog–physical touch initiated by the dog (n = 85), the perception of shared communication (n = 88), and the presence of playful behaviours (n = 100). As a result, these behaviours should be prioritised moving forwards. By incorporating a range of sensors into a robot (e.g., cameras, microphones, and tactile sensors) and supplementing with other technology (e.g., an app, GPS, sensors in the environment), the current state of robotics technology should support the translation of the behaviours mentioned onto a robotic platform [60]. Barriers to implementation remain though, in terms of implementing bonding-specific behaviours (as identified in this study), and a deeper understanding of how social relationships and perceptions develop and change between humans and social robots over time (e.g., [61–63]).

The next step to developing robotic dogs that might begin to fulfil the same roles and bring the same joy as the human—dog relationships surveyed in the present study will be to conduct controlled experiments with people engaging with robotic dogs whose behaviours are modelled on real dog behaviours. By doing this, it should be possible to manipulate the duration and intensity of individual behaviours, as well as establish optimal behavioural boundaries. Through the use of mixed methods approaches, it should further be possible to gain valuable insights into user perceptions and preferences. Conducting further research, to better understand how preferred dog behaviours can (or cannot) be successful modelled onto dog-like robotic systems, stands to greatly inform our understanding of the costs and benefits of dog-like social robots in psychosocial interventions as well as the utility of these machines to serve as long-term social companions for those individuals who are unable to look after a live pet for whatever reason. Additionally, such insights could benefit the development of animal-like avatars (e.g., within apps, virtual reality, or augmented reality), or those working within the arts (e.g., TV or film).

## Conclusion

This study provides detailed insights into dog behaviours perceived as important for human—dog bond formation. The findings should serve as a foundation for further research and development into biomimetic social robots based on dogs. We recommend that next steps focus on exploring the nuances of these behaviours through mixed-methods and testing the applicability and feasibility of programming such behaviours into dog-like robots evaluated in laboratory and real-world settings. Exploring users' reactions and engagement via quantitative and qualitative methods will be important evaluation strategies for ensuring we make progress toward socially useful and likeable companion robots in a responsible manner.

## Supporting information

**S1 Table. Prominent themes identified by coders 1 and 2.**
(PDF)

**S2 Table. Keywords identified within each theme.**
(PDF)

**S1 File. Lexington attachment to pets scale.**
(PDF)

**S2 File. Demographics questionnaire.**
(PDF)

## Author Contributions

**Conceptualization:** Katie A. Riddoch, Roxanne D. Hawkins, Emily S. Cross.

**Data curation:** Katie A. Riddoch.

**Formal analysis:** Katie A. Riddoch.

**Funding acquisition:** Emily S. Cross.

**Investigation:** Katie A. Riddoch.

**Methodology:** Katie A. Riddoch, Roxanne D. Hawkins.

**Project administration:** Emily S. Cross.

**Supervision:** Roxanne D. Hawkins, Emily S. Cross.

**Validation:** Roxanne D. Hawkins, Emily S. Cross.

**Visualization:** Katie A. Riddoch, Emily S. Cross.

**Writing – original draft:** Katie A. Riddoch.

**Writing – review & editing:** Roxanne D. Hawkins, Emily S. Cross.

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
