## [Decision Letter · Decision Letter 0]

9 Feb 2022

PONE-D-21-28840Exploring behaviours perceived as important for human—dog bonding and their translation to a robotic platformPLOS ONE

Dear Dr. Cross,

Thank you for submitting your manuscript to PLOS ONE. After careful consideration, we feel that it has merit but does not fully meet PLOS ONE’s publication criteria as it currently stands. Therefore, we invite you to submit a revised version of the manuscript that addresses the points raised during the review process.

I apologise for the late decision regarding your submission; it was unusually difficult to secure reviewers for this work. I will do my best to speed up the next steps of the procedure.

Below you will find the comments from one independent reviewer, as well as from me as the editor. We are in agreement that the paper presents a relevant piece of work, but that there are a number of issues that need addressing.

We look forward to receiving your revised manuscript.

Kind regards,

I Anna S Olsson, Ph.D.

Academic Editor

PLOS ONE

Journal Requirements:

2. Please include additional information regarding the survey/interview guide used in the study and ensure that you have provided sufficient details that others could replicate the analyses. Specifically, please also include a copy as Supporting Information.

4. We note that Figure 1 in your submission contain copyrighted images. All PLOS content is published under the Creative Commons Attribution License (CC BY 4.0), which means that the manuscript, images, and Supporting Information files will be freely available online, and any third party is permitted to access, download, copy, distribute, and use these materials in any way, even commercially, with proper attribution. For more information, see our copyright guidelines: http://journals.plos.org/plosone/s/licenses-and-copyright.

a. You may seek permission from the original copyright holder of Figure(s) [#] to publish the content specifically under the CC BY 4.0 license. 

6. We note that Figure 2 includes an image of a participant. 

Additional Editor Comments:

Introduction: This section, and in particular lines 67 to 77, presents a very uncritical view of the potential to provide the benefits of pet companionship through robots. The challenges in doing so also need to be presented, and there must be literature addressing this aspect too, which would be appropriate to refer to here.

Methods: You are commended on the adherence to Open Science and the sharing of material through OSF. Some of this information still needs to go in the paper, in particular the description of how participants were recruited. For transparency please also mention how many participants shared their e-mail address.

On lines 150-151 you refer to your original plan to compare the results of those who reported high versus low attachment to their dog. The dog-human relationship has two actors, the dog and the human, and it seems reasonable to assume that the level of attachment would depend on (at least) the characteristics of the dog, the characteristics of the human and the compatibility between these two sets of characteristics. How would your questionnaire allow you to separate the impact of these?

Please be coherent in how you present the themes. They are now presented in one order in Table 1 and in a different order in the narrated part of the Results section, and the terminology of Figure 3 is different for some of the themes.

Line 111 I suggest to reword to “different types of relationships with varying strengths”

Line 208 The phrase “as indicated LAPS questionnaire results” seems to have a word missing.

Line 283 “See Fig 4. For” needs correction

Line 502 “huma-dog” needs correction

Line 557 The phrase “the following” suggests that you will present a list of factors, which is usually preceded by a “:” rather than a “ - “.

Reviewers' comments:

Reviewer's Responses to Questions

**Comments to the Author**

1. Is the manuscript technically sound, and do the data support the conclusions?

Reviewer #1: Partly

2. Has the statistical analysis been performed appropriately and rigorously? 

Reviewer #1: No

3. Have the authors made all data underlying the findings in their manuscript fully available?

Reviewer #1: Yes

4. Is the manuscript presented in an intelligible fashion and written in standard English?

Reviewer #1: No

5. Review Comments to the Author

Reviewer #1: This paper was of interest. However, there are various concerns I have with the quality of the article such as the lack of detail about the questionnaire and its method of distribution and associated time period?.Furthermore, there are various spelling mistakes throughout and there is a lack of discussion of the limitations (i.e., gender bias, self selection bias, sample size and low video usage).

Please see some of the comment below.

Introduction

Line 61 – 64 – “simplicity of modelling non-human animal forms and behaviour.” – I would argue that replication of accurate dog behaviour may not be as simplistic as suggested in some cases e.g. both a happy dog and unhappy dog may wag there tail due to positive or negative arousal etc. I feel there is a fine balance between inaccurate representation based on perceptions/knowledge of dog behaviour versus accurate representation and the importance in accurate replication. I would encourage the author to discuss this as there may be a potential to oversimplify behaviour?

Line 92 – 94 – this may also depend on cultural mileu and vary based on experience etc. see

• https://www.nature.com/articles/s41598-019-52938-4?fbclid=IwAR2cvEHT294UsuPGn8u6_4SL0-mTci1-wqE58FYXIAQ1E7l26muM8P_FPj0

Some article which should be considered.

• Melson, G., Kahn Jr, P., Beck, A., & Friedman, B. (2009). Robotic pets in human lives: Implications for the human-animal bond and for human relationships with personified technologies. Journal of Social Issues, 65(3), 545. http://www.dogica.com/dogpuppy/Robot-Dog-Clone/melson-2009-robotic-pets.pdf

• Ihamäki, P., Heljakka, K. Robot Pets as “Serious Toys”- Activating Social and Emotional Experiences of Elderly People. Inf Syst Front (2021). https://doi.org/10.1007/s10796-021-10175-z

88 – 90 - There is little discussion of robot appearance and its potential effects/importance e.g. the uncanny valley/eeriness, anthropomorphism etc for example: (I appreciate you cant cover everything).

• Walters, M.L., Syrdal, D.S., Dautenhahn, K. et al. Avoiding the uncanny valley: robot appearance, personality and consistency of behavior in an attention-seeking home scenario for a robot companion. Auton Robot 24, 159–178 (2008). https://doi.org/10.1007/s10514-007-9058-3

• Laakasuo, M., Palomäki, J. & Köbis, N. Moral Uncanny Valley: A Robot’s Appearance Moderates How its Decisions are Judged. Int J of Soc Robotics 13, 1679–1688 (2021). https://doi.org/10.1007/s12369-020-00738-6

• Löffler, D., Dörrenbächer, J., & Hassenzahl, M. (2020, March). The uncanny valley effect in zoomorphic robots: The U-shaped relation between animal likeness and likeability. In Proceedings of the 2020 ACM/IEEE international conference on human-robot interaction (pp. 261-270).

• Klüber, K., & Onnasch, L. (2022). Appearance is not everything-Preferred feature combinations for care robots. Computers in Human Behavior, 128, 107128.

Methods

• Good clear ethics statement. But fundamental issues (e.g. was this open to English participants only or all of UK, how long was the questionnaire open for? Distribution method (online but where/how (e.g. social media?)).

• Participant’s data sets? Do you mean respondents or responses?

• Almost half of the sample was removed due to incomplete survey? was this incomplete to the extent that question could not be used. What was the criteria?

• 157 – “questionnaires” and “open questions” – these are the same??

• Good that you have included power analysis – my first question was regarding the low sample, but this is now explained. Was a post hoc power analysis done for the one group?

• 158 – 167 - Check the format of Range for participant factors (e.g. age). I would just state M.

• Participants – demographics and dog experience this maybe easier to displaying as a table.

• 158 – verbal description? Do you mean written?

• 160 – All are dog owners? What about those that are not dog owners or potential owners? E.g. virtual characters what do people look for. If used in social roles individuals may not have previously owned a dog.

• 168 – 170 – this may be better in the result section? Also, I assume the scale was 1 – 5?

• 167 – full demographic details? What else is not provided?

• 172 – followed a link from where – i.e. where was it advertised and how? This is not clear to me?

• 206 – using – should be ‘used’

• 207 – 210 is a repeat of 168 – 170?

• Is the full questionnaire available anywhere? (e.g. OSF)?

Results

• Table 1 – n values? Format (e.g. colour) and the title probably could be more detailed.

Discussion

• Check spelling throughout – e.g. line 502

• This study only covers those who have previously owned dogs?

• There appears limited to no discussion of the major limitations (bias (majority female), small sample and few videos).

6. PLOS authors have the option to publish the peer review history of their article (what does this mean?). If published, this will include your full peer review and any attached files.

Reviewer #1: No

---

## [Author Response · Author response to Decision Letter 0]

19 Jul 2022

Editor Comments:

1. Methods: You are commended on the adherence to Open Science and the sharing of material through OSF. Some of this information still needs to go in the paper, in particular the description of how participants were recruited. For transparency please also mention how many participants shared their e-mail address.

RESPONSE: We thank the Editor for bringing this to our attention. We have now added the following, to address the reviewers’ comments as well: 

• Line 164 – “All participants were recruited by opportunity and snowball sampling over a one-month period, using advertisements posted on various social media platforms.”

• Line 220 – “104 individuals submitted their email address, and a random number generator was used to choose the five winners.”

2. Please be coherent in how you present the themes. They are now presented in one order in Table 1 and in a different order in the narrated part of the Results section.

RESPONSE: We have altered Table 1 so that the themes are listed in order of data coverage (consistent with the rest of the manuscript).

3. The terminology of Figure 3 is different for some of the themes.

RESPONSE: We are further grateful to the Editor for bringing this to our attention. We have now changed the caption to Figure 3 to reflect “independent play”, to be consistent with how this theme is referred to in the main text. We have also changed the text in Table 1 so that the theme names align with the in-text headings (e.g., “Consistency and Predictability” to “Consistency” alone).

4. Line 111 I suggest to reword to “different types of relationships with varying strengths”. 

RESPONSE: We are happy to accept this suggestion and have changed the text to “different types of relationships with varying strengths” (now line 131)

5. Line 208 The phrase “as indicated LAPS questionnaire results” seems to have a word missing. 

RESPONSE: Corrected – now “as indicated by the LAPS questionnaire results” (now line 243).

6. Line 283 “See Fig 4. For” needs correction. 

RESPONSE: Corrected to a lower-case f – “See Fig 4. for images of dogs in close proximity…”

7. Line 502 “huma-dog” needs correction. 

RESPONSE: Corrected to “human—dog”.

8. Line 557 The phrase “the following” suggests that you will present a list of factors, which is usually preceded by a “:” rather than a “ - “. 

RESPONSE: Corrected to “:”.

Reviewer Comments to the Author

Reviewer #1: This paper was of interest. However, there are various concerns I have with the quality of the article such as the lack of detail about the questionnaire and its method of distribution and associated time period? Furthermore, there are various spelling mistakes throughout . Please see some of the comment below.

RESPONSE: We would like to thank Reviewer 1 for this summary. In response to these broad comments, we have done the following:

- Created a sheet documenting the LAPS questionnaire items, and altered in-text (lines 199-202) to include details regarding the measure - “After providing written informed consent, participants completed the Lexington Attachment to Pets Scale (LAPS) – 23 five-point items scored from “Strongly Agree to Strongly Disagree”, developed to measure emotional attachment towards a pet [25]. A full list of items can be found in the S3 Table and on the study’s Open Science Framework (OSF) page, which includes all related study materials - https://osf.io/ycrwh.”

- Created a sheet documenting the demographics questionnaire items. Put the following in-text (lines 204-206), to direct readers to the full list: “Participants were then asked to provide demographic details (e.g., age, gender, length of dog ownership, breed of dog…). A full list of demographics questions is available in S4 Table and the study OSF (https://osf.io/ycrwh).”

- Edited spelling errors within the participant quotes. 

Introduction

Introduction: This section, and in particular lines 67 to 77, presents a very uncritical view of the potential to provide the benefits of pet companionship through robots. The challenges in doing so also need to be presented, and there must be literature addressing this aspect too, which would be appropriate to refer to here.

RESPONSE: We agree with Reviewer 1 that it is imperative that we do not gloss over the ethical (and moral) issues raised by the development of robotic pet companions, and that a more critical view of these issues was lacking in our original submission. We have now added new text to the introduction (lines 87-106) that sketches some of the questions that have legitimately been raised concerning robot development in this space in the past. As Reviewer 1 rightly points out, this is a much bigger debate that extends well beyond the scope of our paper (and indeed, is one that we are delving into in more detail in follow up work), so we have kept our writing on this issue intentionally succinct. The new text reads as follows:

Lines 85-108: “However, it remains imperative to also consider the kinds of social and ethical challenges that the development and introduction of robotic dogs as social companions might bring. In a recent scoping review that evaluated nine studies examining the delivery and impact of interactive pets for older adults (including those with dementia), the authors identified a number of common concerns. These included human users misperceiving robotic pets as living beings (and ethical issues related to deception, even if unintended), ethical issues of attachment, potential negative user reactions, and a number of more practical concerns, such as hygiene and cost (Koh, Ang & Casey, 2021). A number of philosophers have weighed in on the moral and ethical challenges related to the use of robot pets for social companionship, including Robert Sparrow, who voiced concerns over two decades ago that remain strikingly relevant now (Sparrow, 2002). Specifically, Sparrow wrote:

“For an individual to benefit significantly from ownership of a robot pet they must systematically delude themselves regarding the real nature of their relation with the animal. It requires sentimentality of a morally deplorable sort. Indulging in such sentimentality violates a (weak) duty that we have to ourselves to apprehend the world accurately. The design and manufacture of these robots is unethical in so far as it presupposes or encourages this delusion,” (p. 305)

In order to understand the scope and limits of robotic dogs as effective social companions, and indeed, explore the extent to which the general public perceives robotic dogs as ethically questionable or acceptable, researchers may wish to model dog behaviours on appropriate robotic platforms and systematically evaluate the efficacy of these behaviours (compared to, for example, living pets)….”

Line 61 – 64 – “simplicity of modelling non-human animal forms and behaviour.” – I would argue that replication of accurate dog behaviour may not be as simplistic as suggested in some cases e.g. both a happy dog and unhappy dog may wag there tail due to positive or negative arousal etc. I feel there is a fine balance between inaccurate representation based on perceptions/knowledge of dog behaviour versus accurate representation and the importance in accurate replication. I would encourage the author to discuss this as there may be a potential to oversimplify behaviour?

RESPONSE: We have now added sentence (lines 62-66) to clarify that we are referring to relative simplicity of modelling non-human animals opposed to realistic human-like robots (not simplicity in general). This leads on from the previous paragraph which speaks of the high costs and expectations associated with humanoid robots.

“Furthermore, due to the success of human-pet relationships, and the relative simplicity of modelling non-human animal forms and behaviours (compared to human forms and behaviour)…”

Methods

Good clear ethics statement. But fundamental issues (e.g. was this open to English participants only or all of UK, how long was the questionnaire open for? Distribution method (online but where/how (e.g. social media?)).

RESPONSE: We have clarified the details missing, as outlined by Reviewer 1:

• Line 163 added the following: “In total, 283 individuals accessed the online questionnaire (hosted on the Qualtrics survey platform). All participants were recruited by opportunity and snowball sampling over a one-month period, using advertisements posted on various social media platforms”.

• Line 198: specified that the information sheet was in English.

Participant’s data sets? Do you mean respondents or responses?

RESPONSE: Changed the phrase “data” to the following, to aid reader understanding – Line 224: “The open question responses were analysed…”

Almost half of the sample was removed due to incomplete survey? was this incomplete to the extent that question could not be used. What was the criteria?

RESPONSE: We have included a few additional sentences to clarify the exclusion criteria and why such a large proportion was not included. (line 163-169) – 

“In total, the Qualtrics survey platform indicated that 283 individuals started the online questionnaire. All participants were recruited by opportunity and snowball sampling over a one-month period, using advertisements posted on various social media platforms. After the removal of incomplete datasets (that is, those who aborted the study at any point, individuals who left data fields empty, or completely blank surveys), 156 individuals remained. We suspect that there was an issue with the survey platform, as the majority of the incomplete datasets (114 of the 127) contained no data at all”.

• On lines 150-151 you refer to your original plan to compare the results of those who reported high versus low attachment to their dog. The dog-human relationship has two actors, the dog and the human, and it seems reasonable to assume that the level of attachment would depend on (at least) the characteristics of the dog, the characteristics of the human and the compatibility between these two sets of characteristics. How would your questionnaire allow you to separate the impact of these?

RESPONSE: In the paragraph starting 176 we have done the following to address this point (and acknowledge that the current study cannot separate the impact of these):

• emphasise that the attachment is from the human-perspective only – by repeating “those who reported… [level of attachment]”. 

• Add the following data to the OSF, allowing people to scrutinise the characteristics of the relationship if they wish – “A comprehensive demographics table (and the demographics questionnaire) is available on the project’s OSF site - https://osf.io/ycrwh.” (lines 192-193)

In the section “Variation in Human-Dog Relationships” (starting Line 190) we also discuss the point that different dyads have different types of relationships, and there should be greater consideration of compatibility combinations – in future work. We also again reiterate that this study does not have the power to conduct such complex comparisons (608-612).

• 157 – “questionnaires” and “open questions” – these are the same??

RESPONSE: We can appreciate how our previous wording may have confused these terms. On line 144 we rephrase our explanation to the following, to articulate the distinction more clearly - “To address limitations of previous studies, we used open questions (as opposed to closed or fixed-choice questions), to encourage participants to be detailed in their descriptions of the dog behaviours (as opposed to stating abstract or broad qualities briefly).”

We have also changed the wording on line 182, to differentiate between fixed-choice questionnaires and open questions. – “All 153 individuals completed the written aspects of the study (consent, fixed-choice questionnaires, open questions…),”.

• Good that you have included power analysis – my first question was regarding the low sample, but this is now explained. Was a post hoc power analysis done for the one group?

RESPONSE: No, we did not perform a post-hoc power analysis for the one group, based on recommendations regarding the difficulty of interpretation of post-hoc power analyses as outlined by Zhang and colleagues (2019).

Reference: Zhang, Y., Hedo, R., Rivera, A., Rull, R., Richardson, S., & Tu, X. M. (2019). Post hoc power analysis: is it an informative and meaningful analysis?. General psychiatry, 32(4), e100069. https://doi.org/10.1136/gpsych-2019-100069

• 158 – 167 - Check the format of Range for participant factors (e.g. age). I would just state M.

RESPONSE: Changed to the following (avoiding RangeAge formatting) – (MAge = 35.67 years, Range = 21 – 62). Now lines 186-193.

• Participants – demographics and dog experience this maybe easier to displaying as a table.

RESPONSE: In the paper we do not split by dog experience, demographics etc. As a result, we think it would be more appropriate to keep the concise description in the text. We have constructed a table (in the form of an excel file) to convey this information – which is now available on the OSF (https://osf.io/ycrwh). This allows for future research to perform their own data analysis on specific breeds/experience/demographics if they wish.

• 158 – verbal description? Do you mean written?

 RESPONSE: Thank you, yes. Corrected to “written”.

• 168 – 170 – this may be better in the result section? Also, I assume the scale was 1 – 5?

RESPONSE: We have moved this aspect to just before the results section, and altered wording to say the following: “We had originally planned to explore to results of individuals with high vs low general emotional attachment to their dogs - as indicated by the LAPS questionnaire results [25]. This was not possible however, as scores on the questionnaire were close to ceiling (on a scale from 1-5, M = 4.45 and SD = 0.50) - indicating high levels of general emotional attachment across the group.”. Now lines 242-246. 

• 167 – full demographic details? What else is not provided?

RESPONSE: Now line 192. Changed the wording to “A comprehensive demographics table is available on the project’s OSF”. We hope this makes it clear that no information is missing from the manuscript. Instead, the OSF has a more detailed breakdown for each participant.

• 172 – followed a link from where – i.e. where was it advertised and how? This is not clear to me?

RESPONSE: Now line 195. Inserted the following at the start of the procedure (Line 181), to address the lack of clarity – “After gaining ethical approval, researchers posted the research advertisement (containing a link to the survey) on various social media platforms.”

• 206 – using – should be ‘used’

 RESPONSE: Changed to “used”.

• 207 – 210 is a repeat of 168 – 170?

RESPONSE: Thank you for noticing this. We have removed from the former and retained in the latter part of the manuscript.

• Is the full questionnaire available anywhere? (e.g. OSF)?

• RESPONSE: We have created a sheet documenting the LAPS questionnaire items, and altered in-text to include details regarding the measure - “After providing written informed consent, participants completed the Lexington Attachment to Pets Scale (LAPS) – 23 five-point items scored from “Strongly Agree to Strongly Disagree”, developed to measure emotional attachment towards a pet [25]. A full list of items can be found in the S3 Table and on the study OSF - https://osf.io/ycrwh.”. Starting line 199.

• We have further created a sheet documenting the demographics questionnaire items. Put the following in-text, to direct readers to the full list: “Participants were then asked to provide demographic details (e.g., age, gender, length of dog ownership, breed of dog…). A full list of demographics questions is available in S4 Table and the study OSF (https://osf.io/ycrwh).”. Lines 204-206

Results

Format (e.g. colour) and the title probably could be more detailed.

• RESPONSE: We have changed the colour to improve contrast (in line with PLOS guidelines).

• We have Table 1 renamed to “Overview of Key Themes Identified in the Study, and Related Perceptions of Dog Behaviours.”

Table 1 – n values?

RESPONSE: Figure 2 (placed slightly before Table 1) specifies the n values for the themes, and visualises the data coverage (from highest to lowest). Table 1 has been designed to focus on breaking down the behaviours and perceptions, opposed to focussing on the numerical values (which appear in the text body).

Discussion

• Check spelling throughout – e.g. line 502

RESPONSE: Spelling has been triple checked and any mistakes have been corrected in-text, as addressed in a previous comment.

• there is a lack of discussion of the limitations (i.e., gender bias, self selection bias, sample size and low video usage). This study only covers those who have previously owned dogs? All are dog owners? What about those that are not dog owners or potential owners? E.g. virtual characters what do people look for. If used in social roles individuals may not have previously owned a dog.

RESPONSE: We agree that these are important limitations to recognise and discuss. The following section (Lines 553-590) has been added to the discussion to address these concerns, right at the start of the discussion section:

“Study Limitations

Through the use of open questions, this study was able to gather detailed and novel insights regarding dog behaviours perceived as important to human-dog attachment. Due to the lack of video submissions however, there remains a lack of understanding with regards to what the specific behaviours look like, and how they differ between individual dogs, and the different human-dog partnerships. This gap in our knowledge is problematic for those developing animal inspired robots, as it means that we cannot accurately visualise the dog behaviours, nor the physical and technical capabilities are required to facilitate them. To close the gaps in our knowledge, and aid the development of long-term robotic pets and similar, future work should prioritise capturing the nuances of real dog behaviours, and user insights regarding desirable behavioural boundaries. 

Another limitation of the study is that despite conducting this study on a social media platform open to all, the vast majority of respondents identified as female (96%). Although such bias is common in the field of Human Animal Interaction (HAI) work [42], it is a significant issue which negatively impacts the generalisability of the findings. To avoid this problem moving forwards, future studies should tailor their recruitment methods to encourage a diverse and representative sample – for example, by thoughtfully considering the word choice and imagery used in advertisements, and by posting on a range of social media groups (opposed to those dominated by one group). One might also consider monitoring the demographics of incoming responses, and changing tact during the course of recruitment to create improved balance within the sample. 

In this research, a second limitation exists with regards to the generalisability of the results – this time however, by design. Specifically, that all respondents were all individuals who own dogs. On the one hand, this design allowed us to gain insights with depth and breadth, from individuals with extensive experience with dogs – valuable for understanding which behaviours apparently facilitate attachment. On the other hand, this bias means that our insights reflect only the perceptions and preferences of dog owners. In a world where robotic animals will be used by the general public (not just dog owners) this is potentially problematic. By modelling robotic animals on the dog behaviours outlined in this study, developers may create robots which are well-liked by dog owners, but not necessarily the broader public (e.g., individuals who own other types of pets, or people who do not have any experience with animals). To improve the generalisability of the results, and potentially create robotic animals which are accepted on a wide scale, future work should strive to include individuals with a broad range of experience with animals.”

• Line 92 – 94 – this may also depend on cultural mileu and vary based on experience etc. see

• Line 88 – 90 - There is little discussion of robot appearance and its potential effects/importance e.g. the uncanny valley/eeriness, anthropomorphism etc for example: (I appreciate you cant cover everything).

RESPONSE: To briefly address these great points, we have added in the following paragraph (lines 157-190). It is designed to prompt thought about the impact of culture and robot appearance, and provide references for those who wish to read more on those topics. We agree that although we cannot cover everything, these are important points to call out.

“In a world where robotic animals will be used by the general public (not just dog owners) this is potentially problematic. By modelling robotic animals on the dog behaviours outlined in this study, developers may create robots which are well-liked by dog owners, but not necessarily the broader public (e.g., individuals who own other types of pets, or people who do not have any experience with animals). To improve the generalisability of the results, and potentially create robotic animals which are accepted on a wide scale, future work should strive to include individuals with a broad range of experience with animals. On a similar thread, to create robots which are widely used, there should also be increased effort to conduct work examining how context, culture, and form, might influence perceptions of such robots (Cross & Ramsey, 2021; Crowell et al., 2019; Lim, Rooksby & Cross; 2020; Martini, Buzzell & Weise, 2015).”

The following references have been added to the reference list, and incorporated in text, as a result.

Martini, M. C., Buzzell, G. A., & Wiese, E. (2015, October). Agent appearance modulates mind attribution and social attention in human—robot interaction. In International Conference on Social Robotics (pp. 431-439). Springer, Cham.

Crowell, C. R., Deska, J. C., Villano, M., Zenk, J., & Roddy Jr, J. T. (2019). Anthropomorphism of Robots: Study of Appearance and Agency. JMIR human factors, 6(2), e12629.

Lim, V., Rooksby, M., & Cross, E. S. (2021). Social robots on a global stage: establishing a role for culture during human–robot interaction. International Journal of Social Robotics, 13(6), 1307-1333.

Cross, E. S., & Ramsey, R. (2021). Mind meets machine: towards a cognitive science of human–machine interactions. Trends in Cognitive Sciences, 25(3), 200-212.

---

## [Decision Letter · Decision Letter 1]

26 Aug 2022

Exploring behaviors perceived as important for human—dog bonding and their translation to a robotic platform

PONE-D-21-28840R1

Dear Dr. Cross,

We’re pleased to inform you that your manuscript has been judged scientifically suitable for publication and will be formally accepted for publication once it meets all outstanding technical requirements.

Kind regards,

I Anna S Olsson, Ph.D.

Academic Editor

PLOS ONE

Additional Editor Comments (optional):

Reviewers' comments:

Reviewer's Responses to Questions

**Comments to the Author**

1. If the authors have adequately addressed your comments raised in a previous round of review and you feel that this manuscript is now acceptable for publication, you may indicate that here to bypass the “Comments to the Author” section, enter your conflict of interest statement in the “Confidential to Editor” section, and submit your "Accept" recommendation.

Reviewer #1: All comments have been addressed

2. Is the manuscript technically sound, and do the data support the conclusions?

Reviewer #1: Yes

3. Has the statistical analysis been performed appropriately and rigorously? 

Reviewer #1: Yes

4. Have the authors made all data underlying the findings in their manuscript fully available?

Reviewer #1: No

5. Is the manuscript presented in an intelligible fashion and written in standard English?

Reviewer #1: Yes

6. Review Comments to the Author

Reviewer #1: Thank you for the feedback. I am happy with the article in its updated form. I have now accepted the manuscript.

7. PLOS authors have the option to publish the peer review history of their article (what does this mean?). If published, this will include your full peer review and any attached files.

Reviewer #1: No

---

## [Editor Report · Acceptance letter]

1 Sep 2022

PONE-D-21-28840R1 

Exploring behaviours perceived as important for human—dog bonding and their translation to a robotic platform 

Dear Dr. Cross:

I'm pleased to inform you that your manuscript has been deemed suitable for publication in PLOS ONE. Congratulations! Your manuscript is now with our production department. 

Kind regards, 

on behalf of

Dr. I Anna S Olsson 

Academic Editor

PLOS ONE